# Beyond Relevance: Evaluate and Improve Retrievers on Perspective Awareness

**Xinran Zhao**[1], **Tong Chen**[2], **Sihao Chen**[3], **Hongming Zhang**[4], **Tongshuang Wu**[1]
[1]Carnegie Mellon University, [2]University of Washington, [3]University of Pennsylvania,
[4]Tencent AI Lab, Bellevue
{xinranz3, sherryw}@andrew.cmu.edu

## Abstract

The task of Information Retrieval (IR) requires a system to identify relevant documents based on users' information needs. In real-world scenarios, retrievers are expected to not only rely on the semantic relevance between the documents and the queries but also recognize the nuanced intents or *perspectives* behind a user query. For example, when asked to verify a claim, a retrieval system is expected to identify evidence from both supporting vs. contradicting perspectives, for the downstream system to make a fair judgment call. In this work, we study whether retrievers can recognize and respond to different perspectives of the queries — beyond finding relevant documents for a claim, can retrievers distinguish supporting vs. opposing documents? We reform and extend six existing tasks to create a benchmark for retrieval, where we have diverse perspectives described in free-form text, besides root, neutral queries. We show that current retrievers covered in our experiments have limited awareness of subtly different perspectives in queries and can also be biased toward certain perspectives. Motivated by the observation, we further explore the potential to leverage geometric features of retriever representation space to improve the perspective aware­ness of retrievers in a zero-shot manner. We demonstrate the efficiency and effectiveness of our projection-based methods on the same set of tasks. Fur­ther analysis also shows how perspective awareness improves performance on various downstream tasks, with 4.2% higher accuracy on AmbigQA and 29.9% more correlation with designated viewpoints on essay writing, compared to non-perspective-aware baselines[1].

## 1 Introduction

Large Language models (LLMs) have now reshaped various real-world applications, be it creative writing (Yuan et al., 2022), natural-language search (Ziems et al., 2023), embodied agents (Wang et al., 2023), or web-based AI assistance (Yao et al., 2023; Xie et al., 2024). However, there is a widely recognized limitation affecting these applications (Khattab & Zaharia, 2020): the knowledge embedded within the parameters of language models can often be opaque, static, and inefficient. To enhance the practical usability of LLMs for knowledge-intensive tasks, substantial efforts have been dedicated to retrieval-augmented generation (RAG, Lewis et al., 2020) — to fetch external knowledge using *retrievers*, and thereby augment LLMs as part of the context.

Retrievers, however, are double-edged swords. While they bring in richer contexts, they can also become bottlenecks when they miss the nuanced demands of users. For instance, consider the scenario in Figure 1 where a user requests an LLM to compose an essay *opposing* the claim, "African governments should enforce stricter animal protection measures." It is crucial for the retriever to accurately grasp the user's stance of opposition; A failure might result in the inclusion of documents that, while relevant, do not align with the user's

---

[1]Our code is available at https://github.com/colinzhaoust/pir.

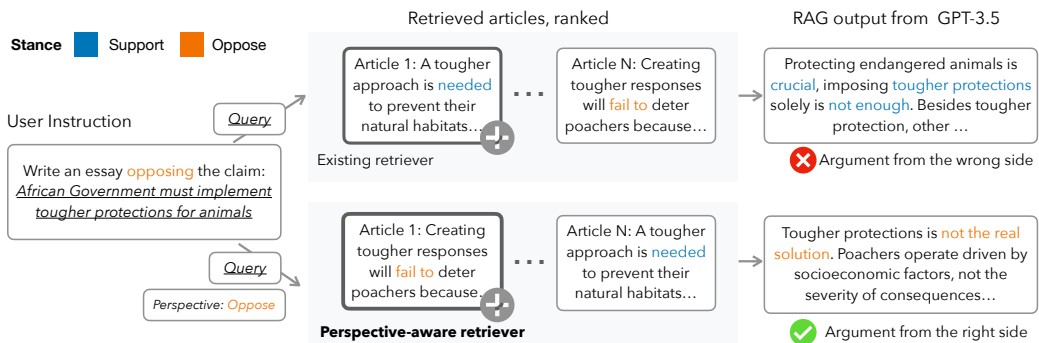

Figure 1: An example of how perspective-ware information retrieval differs from the current retrieval pipeline. Perspectives further specifying the intent, e.g., *"Article that opposes"*, will influence the ranks of relevant articles, hence influencing the downstream task performance.

*perspective* (e.g., those that actually *support* the claim). Such misalignment could lead to a lack of helpful information or even *Ripple Effects* (Cohen et al., 2023), which skew the whole LLM's output in a biased manner. However, existing RAG systems typically leverage fuzzy retrievers (Cross, 1994) that emphasize the *overall* lexical or semantic similarity, e.g., token-based BM25 (Robertson & Zaragoza, 2009) or embedding-based DPR (Karpukhin et al., 2020). It remains unclear whether retrievers can adequately capture the subtle perspective differences — In Figure 1, the perspective is only reflected by a single keyword "opposing."

In this work, we **evaluate and improve retrievers on their sensitivity to the subtly different perspectives in queries**. First, to enable systematic analysis of retrievers' perspective awareness, we create a Perspective-aware Information Retrieval benchmark (**PIR**). Each instance in PIR is a tuple of {base query (e.g., "African government..."), perspective (e.g., support/oppose), gold documents}. We create PIR by repurposing six existing datasets[2] in various domains, such that the benchmark covers multiple real-world tasks that require retrieval (e.g., news, argument mining, question answering, etc), as well as diverse perspectives (left or right-wing ideologies for news, supporting or refuting claims for fact-checking, etc.). In total, we collect 7,494 diverse queries and 10,286 corpus candidates. Along with the data structure, we further design a new metric, *p*-recall, for capturing if retrievers perform consistently across perspectives.

We test five off-the-shelf retrievers using the dataset. We find that existing retrievers are biased towards choosing candidates from certain perspectives and struggle to distinguish between semantically similar queries that convey different perspectives (Section 2.4). In fact, PIR also exposes potential social bias within these systems. For instance, we observe a tendency for retrievers to favor news sources from specific countries, regardless of explicit instructions in the queries to target other countries (Section 2.5).

Then, to improve perspective awareness, we propose Perspective-aware Projection (**PAP**), i.e., to emphasize the perspectives, by projecting the embeddings of the queries and corpus candidates onto the perspective through vector space computation. We show that PAP outperforms other baselines in various settings (Section 3.2), offering a straightforward and efficient way to improve the perspective sensitivity of the retrieval process without needing additional fine-tuning.

Moreover, perspective-aware retrieval can significantly improve downstream task performance (Section 3.3). With PAP, state-of-the-art LLMs like GPT-3.5-Turbo (OpenAI, 2022) answer ambiguous questions with higher accuracy (4.2-point higher on AmbigQA (Min et al., 2020)); Similarly, in writing tasks like Figure 1, models can generate essays that more closely align with the designated viewpoints (29.9% more correlation than baseline).

In summary, our main contributions are:

---

[2]The proposed pipeline can be generalized to all domains with similar-formatted sources.

- We introduce a novel benchmark, PIR, to study how retrievers are aware of the intrinsic perspective in the queries in various real-world tasks reflecting users' intent.
- We design a projection-based method, PAP, to improve the perspective awareness of off-the-shelf retrievers, which requires no fine-tuning and is with a minimum adaptation of the original pipeline.
- Further analysis of PIR reveals how bias exists in retrieval results and how downstream RAG performance can be improved with perspective-aware retrievers.

## 2 Perspective-aware Information Retrieval

### 2.1 Motivation and Task formation

Recently, various retrieval methods have shown impressive performance in finding semantic relevance. However, one issue is that the relevant document extracted may be biased towards certain perspectives and do not match the actual user need in the query. For example, in a scenario where a user retrieves evidence (by *find an article about topic X*) to write an article *against* a certain topic, with on average 58.5 percent chance, the top selected document will be on the *supporting* side[3]. Such observation shows that retrieval action solely based on semantic relevance may limit the diversity and effectiveness of the retrieval.

To tackle this issue, we identify an intrinsic dimension in addition to the query: the *perspective*. Perspectives are **further specifications of users' actual needs** (e.g., find an article that supports/opposes) attached to the queries (e.g., the African government must implement tougher protections for animals). These specifications naturally exist explicitly or implicitly in queries seeking information. They also tend to be conveyed by relatively short phrases compared to the whole query: One- or two-word differences may trigger contrastive perspectives and hence different target documents in the corpus.

We formally define the perspective-aware information retrieval (PIR) task as follows. We denote the set of retrieval candidates (corpus) as $\mathcal{C}$, and each document or passage in the corpus as $\mathbf{c} \in \mathcal{C}$. Given a query $q$ and target perspective $p$, the goal is finding $\mathbf{c} \in \mathcal{C}$ such that $\mathbf{q}$ and $\mathbf{c}$ are similar from the perspective of $p$. A perspective-aware retriever, $\phi$, takes the $\mathbf{q}$, $\mathbf{p}$, and $c$ as the input and output a similarity score $\phi(\mathbf{q}, \mathbf{p}, \mathbf{c})$. $\mathcal{C}$ is then ranked based on the similarity scores of each candidate. In the rest of the paper, we also denote each query with perspectives deprived as root query $r$. Typically, $\mathbf{q} = \mathbf{p} + \mathbf{r}$. For each $\mathbf{r}$, we sample at least two $\mathbf{p}$, with corresponding gold $\mathbf{c}$, attached to it in the dataset, for the purpose of conducting contrastive experiments and evaluation. Notice that in the realistic retrieval scenarios, only $\mathbf{q}$ is assumed to be observed, in Section 3.4, we explore the automatic deprivation of $\mathbf{p}$.

### 2.2 Datasets

Existing retrieval benchmarks usually do not explicitly consider perspectives. However, empirically, we find that a lot of datasets in domains that need retrieval can be easily repurposed for testing perspective awareness. For example, AllSides (Baly et al., 2020) is a dataset originally designed for *ideology* classification on news articles, where each news also has the metadata of its *topic*. We can easily create queries like "Find news articles that are about [topic]" which specifies a root query for retrieval; We can further add "...from [ideology]" to specify the *ideology* perspective. Through such transformation, we can create pairs of queries that differ in a single perspective (e.g., "Find news articles that's about [topic] from [Left- vs. Right-wing ideology]").

**Data source.** Inspired by this observation, we build the PIR benchmark by reformatting six existing datasets, as summarized in Table 1. Similar to the popular retrieval benchmark BEIR (Thakur et al., 2021), we cover **diverse** retrieval domains, including Argument, News, Question-Answering (QA), and Fact-check. As shown in the table, the data also spread across diverse topics and formats (in terms of query and corpus length).

---

[3]Detailed analysis is in Section 2.5 after we introduce the data creation.

| Domain | Perspectives (underline) | Source Dataset | Query # | Query Len | Corpus # | Corpus Len |
|---|---|---|---|---|---|---|
| Argument | Stance | Perspectrum (Chen et al., 2019) | 1,431 | 15.1 | 2,773 | 11.0 |

*Ex.:* Find a claim that supports / opposes the argument: *It should be allowed to have military recruitment in schools.*

| QA evidence | Question-Specific | AmbigQA (Min et al., 2020) | 864 | 12.8 | 864 | 31.5 |

*Ex.: What is the legal age of marriage, without parental consent or other authorization* in Nebraska / all but two states in the USA?

| News | Political ideology | AllSides (Baly et al., 2020) | 645 | 12.4 | 645 | 1,089.2 |

*Ex.:* From Left-wing / Right-wing media, *find a news article on the topic: terrorism*

| Fact-check | Evidence Type | Ex-FEVER (Ma et al., 2023) | 1,104 | 44.3 | 1,104 | 27.7 |

*Ex.:* Find a claim that this sentence refutes / supports: *Mother Teresa was born in Macedonia, a country in Europe where the residents are called Macedonians. Macedonian is a Slavic language.*

| News | Topic, Location | AGNews (Yu et al., 2023b) | 1,450 | 87.4 | 2,900 | 162.2 |

*Ex.:* Find a news article that happened in Canada / Brazil and has the same topic as: *An investigative analysis has found that India's lack of investment in public health infrastructure...*

| Story | Similarity Type | StoryAnalogy (Jiayang et al., 2023) | 2,000 | 26.0 | 2,000 | 16.2 |

*Ex.:* Find a story that has similar entities with / is an analogy to: *Fertilize the soil. Mix seeds into the fertilized soil.*

Table 1: Statistics and examples of the selected reformatted datasets. Query/Corpus #/Len denotes the sizes and average number of words for the queries and corpora, respectively.

**Data format.** We unify all the datasets, such that each task (i.e., domain + perspective) contains the following components: (1) queries: a piece of text containing a question or a sentence describing the need; (2) a corpus: a list of candidates (documents or passages) that contain potentially useful information for queries of the same task; (3) a key reference: the mapping between queries and corpus candidates.

Most importantly, we define each query to explicitly contain a *root query* (italicized in the examples in Table 1), and a *perspective* (underlined). To evaluate **perspective sensitivity**, we make sure that for each *root query*, there are at least two queries that have *different perspectives* on each root query. To enable unambiguous evaluation of perspective awareness, we use heuristics to ensure that all the queries with perspectives have **mutually exclusive** matches to golden retrieval documents. For example, (Yu et al., 2023b) carefully curates an AGNews-style dataset with one utterance per feature combination (e.g., length, style, and source, etc.) Moreover, we carefully constructed the data transformation function, so as to make the synthesized queries are **natural sounding**. More details on how we reformat and extend the source datasets are in Section A.2.

## 2.3 Evaluation Metric

To capture the retriever awareness of perspectives, we adjust the standard metric *Recall* into a new metric denoted as *p-Recall*: For each root query, we collect the retrieval performance of all perspectives and average them. The overall performance will then be measured by the micro-average across all root queries, with the consideration of consistent perspective awareness.

Adopting the notations in Section 2.1, with a retriever $\phi$, a corpus $\mathcal{C}$, root queries $r \in \mathcal{Q}_{\text{root}}$, per query perspectives $p \in \mathcal{P}_q$, and a threshold $k$, the metric is computed as:

$$p\text{-Recall@}k = \frac{1}{|\mathcal{Q}_{\text{r}}|} \sum_{\mathbf{r} \in \mathcal{Q}_{\text{r}}} E_{\mathbf{p} \in \mathcal{P}_q}[\text{success}(\phi, \mathbf{p}, \mathbf{q}, \mathcal{C}, \mathbf{k})], \mathbf{q} = \mathbf{p} + \mathbf{r}, \tag{1}$$

where success(.) denotes if the gold document of query $\mathbf{q}$ with perspective $\mathbf{p}$ appears in the top $k$ in corpus $C$, ranked by $\phi$, similar to Recall@$k$.

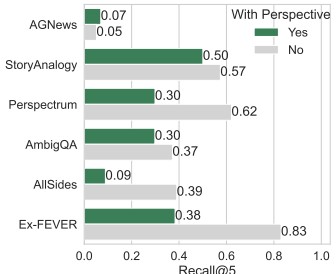

Figure 2: Retrieval performance (Recall@5) of queries with or without perspectives, macro-averaged over all the retrievers.

| | BM25 | DPR | SimCSE-sup | SimCSE-unsup | Contriever | TART |
|---|---|---|---|---|---|---|
| AGNews | 9.1 | 8.0 | **11.0** | 10.4 | 3.5 | 1.0 |
| StoryAnalogy | 71.3 | 45.3 | **86.1** | 76.4 | 9.1 | 12.3 |
| Perspectrum | 39.1 | 35.3 | **52.1** | 50.0 | 10.4 | 9.2 |
| AmbigQA | 23.8 | **55.9** | 53.3 | 48.1 | 1.5 | 0.8 |
| AllSides | **17.6** | 7.8 | 10.8 | 13.0 | 2.8 | 1.7 |
| Ex-FEVER | **53.7** | 47.1 | 52.4 | 53.0 | 1.9 | 21.8 |
| Avg. | 35.8 | 33.2 | **44.3** | 41.8 | 4.9 | 7.8 |

Table 2: Perspective-aware performance of different retrievers on PIR, with $p$-Recall@5. as the metric. Avg. denotes the macro-average performance across the tasks. Best-performing entries for each row are **bolded**.

## 2.4 Experiment: Measure Retrievers' Perspective Awareness

**Baseline Retrievers**    Similar to Thakur et al. (2021), we present PIR as an evaluation benchmark for zero-shot information retrieval. We test the following zero-shot baselines to reveal the perspective awareness of different kinds of retrieval systems. We consider retrievers with various design: token-based similarity (BM25, Trotman et al., 2014), dense embeddings fine-tuned on retrieval tasks (DPR, Karpukhin et al., 2020), sentence embeddings (SimCSE, Gao et al., 2021), embeddings learned through unsupervised contrastive learning (Contriever Izacard et al., 2022), and instruction-fine-tuned embeddings (TART, Asai et al., 2022).

For SimCSE, we consider both the unsupervised (-unsup) and supervised (-sup) versions. For TART, following (Oh et al., 2024), we use TART-dual that is based on Contriever. We introduce details of these baseline retrievers in Section A.3 in the appendix.

**Results**    We first compare retriever performances on queries with and without perspectives, using the standard metric Recall@5. As shown in Figure 2, while retrievers can distinguish the semantically different queries (i.e., reasonable no perspective performance), they generally **display insufficient perspective awareness** — In some extreme cases, they only achieve half the performance when required to include perspectives. This could result from retrievers over-emphasizing the overall semantic similarity while overlooking specific trigger words for perspectives. The observation highlights the usefulness of PIR for studying intrinsic perspective awareness.

One exception is on AGNews, where retrievers perform slightly better when retrieving perspectives (though still very poor). As further analyzed below in Section 2.5, retrievers inherently are biased towards News Articles from certain countries (e.g., Brazil); In such cases, more explicitly querying for less preferred countries might have counterbalanced the bias. More details are in Section A.5.

Moreover, we find that **PIR can help effectively compare retrievers.** In Table 2, we present per-retriever performance with our defined metric $p$-Recall@5. We observe that:

- BM25 could perform better than other dense retrievers when it is appropriate for the task (echoing findings in Thakur et al. (2021)).
- Sentence embedding methods perform well. SimCSE, which is not specifically trained with retrieval tasks, performs the best among all retrievers. One reason behind this can be that sentence embeddings retain more semantic understanding than DPR, which helps identify the perspectives. Among all retrievers, SimCSE-sup, which is further fine-tuned on NLI tasks (not retrieval tasks), performs the best. This may reveal the innate connection between retrieval and natural language inference.
- Domain-specific retrievers like DPR might have limited generalizability, echoing Thakur et al. (2021). Token-based BM25 performs better than DPR except for AmbigQA, a dataset that has questions on Wikipedia paragraphs which DPR is trained on.

| Retriever | Perspectrum | | AllSides | | |
|---|---|---|---|---|---|
| | Support | Undermine | Left | Right | Center |
| BM25 | 55.6 | 44.4 | 27.9 | 44.3 | 27.9 |
| DPR | 55.5 | 44.5 | 29.4 | 37.6 | 32.9 |
| SimCSE-sup | 58.8 | 41.2 | 32.0 | 41.2 | 26.8 |
| SimCSE-unsup | 59.2 | 40.8 | 26.1 | 39.6 | 34.2 |
| Contriever | 66.1 | 33.9 | 28.6 | 42.9 | 28.6 |
| TART | 55.6 | 51.9 | 33.3 | 66.7 | 0.0 |

Table 3: Portion of different perspectives when the retrievers successfully retrieve relevant documents with root queries. We can observe that retrievers are biased towards supporting documents (for Perspectrum) or news articles from the right-wing media (for AllSides). In the corpus, the number of entries related to each perspective is designed to be equal.

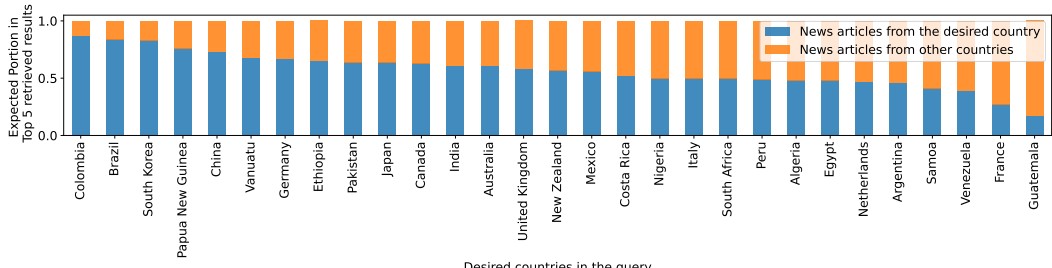

Figure 3: Expected portion of news articles from the desired or other countries in the top 5 retrieval results with SimCSE-sup on AGNews. Queries are from the location perspective, e.g., Find a news article on X topic and happen in Y, where Y is the desired country. We can observe that retrievers show imbalanced performance across countries. For example, users seeking news from Guatemala will experience a lower chance of satisfied retrieval than from Colombia. In the corpus, the numbers of articles per country are designed to be equal.

- Current instruction tuning may not fully solve the sensitive perspective change issue. Though TART outperforms its base: Contriever, both perform worse than others.

## 2.5 Analysis: Biases in Retrievers Revealed by PIR

In Section 2.1, we briefly described the potential bias of retriever results. With PIR, we can now formalize the analysis with quantitative observations. We highlight two

**Retrievers can present biased opinions.** We conduct retrieval on PIR with perspective-neutral *root queries*, to answer the question: Are retrievers inherently biased towards documents with certain perspectives? As an example, we run experiments on Perspectrum and AllSides as they have a fixed set of perspective options (Stance: support vs. oppose; Political ideology: Left, Right, Center). From Table 3, we can observe that, even though the corpus candidates are balanced with respect to perspective by design, retrievers are still biased towards extracting Supporting documents or right-wing news, which denotes that retrieval without perspectives may limit the result comprehensiveness.

**Retrievers are not equally usable across countries.** In our experiments on AGNews, we notice that news from some countries are more easily retrieved than others. For example, in Figure 3. Users who seek news articles from Colombia and Brazil have a higher chance (around 80%) of finding the desired articles in the top 5 retrieved results, compared to those who seek news articles from France or Guatemala (around 20%). We present further analysis on (1) the bias when the desired countries are not in the queries; and (2) what the other countries are, in Figure 3 in Section A.10 in the appendix.

# 3 Improving Perspective Awareness

In this section, we explore improving the perspective awareness upon the best-performing retriever in Section 2.4: SimCSE. We hypothesize that one reason behind the lack of perspective awareness is that the perspectives are over-shaded by the overall query semantics since different perspectives are commonly conveyed by only one or two words. Accordingly, we propose to enhance awareness by projecting the query representation to the hyperplane of the perspectives, so that the similarity between the query and candidates *is forced to be conditioned on the perspectives*.

Here, we assume that the perspectives in the queries are pre-extracted with heuristics. We explore the extended setting to extract perspectives from raw queries with a generative model in Section 3.4.

## 3.1 Perspective-aware Projection (PAP)

Our method first **encode** each query, root query, perspective, and corpus candidate into embeddings $\mathbf{q}$, $\mathbf{r}$, $\mathbf{p}$, and $\mathbf{c}$ respectively, with language models (e.g., SimCSE). We treat all these elements as raw textual sentences and use the mean of token embeddings as sentence embedding. Theoretically, any transformer-based model could be used as the encoder, but by default, we choose to use SimCSE as we see which reflects its effective sentence embedding capability.

Then, we perform perspective emphasis, by **projecting** the query embedding $q$ and (optionally) corpus instance $c$ onto the hyperplane that is vertical to the perspective embedding $(p)$, via:

$$\mathbf{q}_p = \mathbf{q} - \frac{\mathbf{q} \cdot \mathbf{p}}{\|\mathbf{p}\|^2}\mathbf{p}, \quad \mathbf{c}_p = \mathbf{c} - \frac{\mathbf{c} \cdot \mathbf{p}}{\|\mathbf{p}\|^2}\mathbf{p}.$$

Such that we can retrieve relevant instances via the cosine distance between the projected $\mathbf{q}_p$ and $\mathbf{c}_p$. Corpus projection is arguably more expensive, which we make to be optional. We denote the vanilla and efficient version without corpus projection as **PAP**, and the one with corpus projection as **PAP+**. It is worth noting that there can be a tunable hyperparameter added before the projected component of $q$ or $c$ to control the level of perspective emphasis.

**Computational Efficiency.** In practice, we would hope to implement both PAP variants through a combination of offline preparation and online inference. Following the RAG convention, we will always pre-compute all the corpus instance embeddings $\mathbf{c} \in \mathcal{C}$. In real scenarios, for tasks with known desired perspectives (e.g., supporting and opposing for argument retrieval), we can also pre-compute the projection of corpus entries $\mathbf{c}_p \in \mathcal{C}_p$ to improve efficiency. $q$ and $q_p$, on the other hand, have to be computed on the fly. To speed up the inference, we conduct the projection of all candidates at once with a matrix operation and retrieve the most relevant one with a maximum inner product search (MIPS) operation. Empirically, we could use existing MIPS algorithms such as FAISS (Johnson et al., 2019) to reduce the overall inference complexity to $log(\|\mathcal{C}\|)$.

## 3.2 Experiment: Retrieval Performance

We compare the projection-based PAP with alternative interventions at the vector level for including perspectives, as summarized in Table 4. We consider baselines as operations on both the embedding space level (e.g., *add*, *concat.*, etc) and similarity score level (*dual-sum* and *tri-sum*) [4]. From the table, we can observe that, although various methods designed all help with improving $p$-Recall@5, our proposed PAP performs the best for both settings with or without corpus embedding modification. While we set to improve on SimCSE, we observe that PAP can robustly work with various backbones and scales. We include more details in Section A.7. Though the improvement might seem marginal, we show below that it makes a significant impact on the downstream task performances.

---

[4]We use cosine similarity to keep consistent with the common training pipeline of various retrievers. Recently, Steck et al. (2024) discuss the use of other similarity functions

| Method | Scoring function per query | Description | Avg. |
|---|---|---|---|
| baseline | $\cos\angle(\mathbf{q}, \mathbf{c})$ | original best performance in Table 2 | 44.3 |
| add | $\cos\angle(\mathbf{r} + \mathbf{p}, \mathbf{c})$ | encode the root and perspectives separately, use the vector sum for query | 45.8 |
| concat. | $\cos\angle((\mathbf{r}, \mathbf{p}), \mathbf{c})$ | encode the root and perspectives separately, using the vector concatenation for query | 45.4 |
| cast | $\cos\angle((\mathbf{q} - \mathbf{p}, \mathbf{c})$ | compare the query and corpus entries with the query perspective subtracted | 43.7 |
| cast+ | $\cos\angle((\mathbf{q} - \mathbf{p}, \mathbf{c} - \mathbf{p})$ | compare the query and corpus entries with the perspective subtracted | 46.0 |
| dual-sum | $\cos\angle((\mathbf{r}, \mathbf{c}) + \cos\angle(\mathbf{p}, \mathbf{c})$ | consider separate score for root and perspectives | 45.0 |
| tri-sum | $\cos\angle((\mathbf{r}, \mathbf{c}) + \cos\angle(\mathbf{p}, \mathbf{c}) + \cos\angle(\mathbf{q}, \mathbf{c})$ | consider separate score for root, perspectives, and queries | 45.4 |
| PAP | $\cos\angle(\mathbf{q}_p, \mathbf{c})$ | project the query to the perspective plane | 46.0 |
| PAP+ | $\cos\angle(\mathbf{q}_p, \mathbf{c}_p)$ | project both query and corpus entries to the perspective plane | **46.4** |

Table 4: Performance and descriptions of different methods to improve the retriever perspective awareness. + denotes that the method contains modification over the corpus vectors. $r, p, c, q$ denote the embeddings of corresponding root queries, perspectives, corpus entries, and queries, respectively. $\cos\angle(.)$ denotes the cosine similarity. Details about the proposed PAP are in Section 3.1. Avg. denotes the average performance over the six tasks of PIR, with $p$-Recall@5 as the metric.

## 3.3 Analysis: Impact on Downstream Tasks

To uncover the impact of PAP on downstream tasks, we study retrieval-augmented generation (RAG) performance with AmbigQA (120 examples, with top-1 retrieved document) and Perspectrum (30 examples, with top-5 retrieved documents). We use GPT-3.5-Turbo (OpenAI, 2022) as the reader model.

For AmbigQA, we define the downstream task to be the original question-answering task. For example, for the question: *What is the legal age of marriage, without parental consent or other authorization, in Mississippi?* the answer is: 21. We use the exact match between answer and output as our metric and report the overall accuracy. For Perspectrum, we design an argumentative essay writing task. Specifically, we ask models to write a short essay based on the topics and corresponding stances (derived from queries in our Perspectrum task), e.g., *Topic: It should be allowed to have military recruitment in schools; Stance: Oppose*. We measure if the essays written follow the sentiment of the topic-stance pairs by computing the Pearson correlation between their sentiment polarity. We use the polarity scores extracted by TextBlob (Loria, 2020). Detailed experiment settings and prompts are presented in Section A.8. We also identify and call for human evaluation and other perspective-related downstream tasks as important future work, for example, retrieving different lemma variants in TheoremQA (Chen et al., 2023b).

We set no retrieval (remove the knowledge part in the prompt) and gold retrieval (include gold knowledge attached to the question) as baselines, and compare our proposed PAP, PAP+ with their backbone vanilla retriever SimCSE-sup (pre-pend the retrieved results as background knowledge before questions). From Table 5, we observe that retrievers with perspective awareness lead to improved performance on the downstream task, besides the retrieval performance shown in Table 4. For Perspectrum, without perspective-aware retrieval, involving controversial documents as the background may lead to output essays that do not follow the original instructions.

| Retriever | AmbigQA (acc.) | Perspectrum (corr.) |
|---|---|---|
| no | 62.5 | 16.1* |
| gold | 77.5 | 38.7* |
| SimCSE | 72.5 | -2.4 |
| PAP | 74.2 | 18.5* |
| PAP+ | **76.7** | **20.9*** |

Table 5: Performance of different retrievers on AmbigQA and Perspectrum. acc./corr. denote accuracy and Pearson correlation coefficient, respectively. Both metrics are the higher the better. We use * to denote entries with p-value $< 0.05$.

Why is the downstream task so much better, when $p-$recall@5 only shows marginal improvement? Similar to Lee et al. (2021), we speculate that the current metric of retrieval performance may not necessarily correlate with the end task performance: Metrics like recall only measure *if the correct document is retrieved*, but do not capture how incorrect other involved documents might be. If top-5 documents are included but four of them are relevant but incorrect, the model can easily get misled (similar to Figure 1). This aligns with the

findings from Yu et al. (2023a): even if the right answer is in the context, models can also potentially output the wrong answer that is also in the context. Aligning the retrieval and downstream task metrics is important but out of the scope of this paper.

### 3.4 Feasibility Check: Generative Perspectives from Queries

In this paper, we discuss how perspectives naturally exist in user queries, and conduct experiments assuming known perspectives on various tasks. However, besides the heuristics, can we extract perspectives automatically?

We explore whether it would be practical to generalize the PIR setting to the real world, by trying to acquire meaningful generative perspectives with LLMs. Specifically, we extract the perspectives from queries with GPT-3.5-Turbo in a zero-shot setting. We select 50 examples from each task and test if the perspectives extracted match the annotated perspectives. We prompt the model with instructions on extracting part of the sentence specifying perspectives. Following Zhao et al. (2023), we measure the similarity by a common metric QA-F1 used in open-domain QA, calculating the max uni-gram overlap between two pieces of text. We present the details for the prompt and metric computation in Section A.8. The prompt is simple and not heavily engineered to test if models can distinguish user meta-description (e.g., Find a news article) and general information (e.g., the source news content) well. Besides the overlap, we also measure if the performance is consistent between annotated and generated perspectives. We use Recall@5 as the metric to measure the retrieval performance changes on each task, with the corresponding full corpus.

| Dataset | QA-F1 | Δ BM25 | Δ DPR | Δ SimCSE |
|---|---|---|---|---|
| AGNews | 78.0 | 12-12 | 4-4 | 8-8 |
| StoryAnalogy | 65.9 | 56-60 | 26-32 | 36-38 |
| Perspectrum | 68.6 | 26-28 | 38-38 | 50-50 |
| AmbigQA | 70.2 | 21-18 | 46-44 | 44-44 |
| AllSides | 66.4 | 6-8 | 8-10 | 8-8 |
| Ex-FEVER | 68.6 | 80-82 | 60-60 | 72-72 |

Table 6: Performance of automatic perspective extraction (QA-F1) from queries on different tasks with GPT-3.5-Turbo, with corresponding retrieval performance changes (Recall@5). A-B denotes the performance with annotated perspectives (A) and generated perspectives (B).

From Table 6, we can observe that GPT-3.5-Turbo achieves good performance in extracting the perspectives from queries. On the one hand, good performance denotes the naturalness of decomposing queries with perspectives and root queries. On the other hand, since perspective extraction can be done automatically, our proposed PAP can potentially be extended in general scenarios where can not rely on heuristics to find perspectives. Regarding the retrieval performance, we can observe that the retrieval performances from annotated and generative perspective extraction are similar across tasks and retrievers, which validates our claim of the potential to automate perspective extraction in real scenarios.

## 4 Related Work

**Task-aware Retrieval**   Recent neural retrievers (Karpukhin et al., 2020; Santhanam et al., 2022) have shown their superiority over lexical retrievers (e.g., BM25, Robertson & Zaragoza, 2009; Trotman et al., 2014) in various domains. Due to the high cost of annotating retrieval datasets for new target tasks, there is an increasing focus on improving the generalizability of neural retrievers across diverse sets of domains and tasks (e.g., BEIR, Thakur et al., 2021). Retrievers trained with unsupervised optimization targets (Izacard et al., 2022) or large-scale datasets (Ni et al., 2022; Wang et al., 2022) show surprisingly good generalization ability across diverse domains.

Given improved cross-domain generalization of neural retrievers, studies have started to investigate the cross-task generalization (Asai et al., 2022; Su et al., 2022). Inspired by the promise of instruction tuning on language models, training retrievers with task-specific instructions achieves improved in-domain performance with an instruction-prefix (Yang et al., 2021; Mysore et al., 2022; Ravfogel et al., 2023; Zhang et al., 2023; Jeong et al., 2024) or

an instruction-module in REMOP (Liang et al., 2023). Instruction-tuning also demonstrates zero-shot generalization on unseen domains and tasks (e.g., TART, Asai et al., 2022). In our work, we present TART still can not solve the perspective imbalance. Yet perspective-aware instruction fine-tuning can be one important future direction to reduce it.

**Retrieval Evaluation** The benchmarks designed for evaluating generalization ability across tasks include BEIR (Thakur et al., 2021), LoTTE (Santhanam et al., 2022), and X2 (Asai et al., 2022). In these datasets, search queries vary across different tasks. Therefore, we cannot compare the rank difference on the same query given different tasks (or perspectives). Additionally, M-BEIR (Wei et al., 2023) provides diverse retrieval tasks but is constrained in our study by its multi-modal nature.

Some concurrent works also evaluate the retriever's instruction following ability given instructions in natural language form, while all these works define instruction and tasks from different views. InstructIR (Oh et al., 2024) focuses on user preferences given instructions that contain the user's characteristics such as background, situation, job, hobbies, etc. FollowIR (Weller et al., 2024) constructs instructions to explicitly describe what documents are relevant and not relevant. In our work, we motivate intent perspectives from various end-tasks that users may need. We construct tasks by seeking different ways to measure the similarity between a query and a document, inspired by prior works on conditional semantic similarity (Deshpande et al., 2023). Despite the different definitions of instruction and tasks, these studies draw similar conclusions that the existing neural retrievers are struggling to follow the instructions.

**Conditional Similarity and Text Embeddings** Besides the conditional query-document relevance discussed above, recently, there has been a discussion regarding conditional text embeddings in the community. MTEB (Muennighoff et al., 2023) is proposed to evaluate text embedding performance on diverse tasks. C-STS (Deshpande et al., 2023) highlights how short textual description will influence similarity between sentence pairs. (Su et al., 2022) presents how involving instructions improves text embedding performance on corresponding tasks. In our work, we introduce perspectives as a factor that influences the query-document relevance.

## 5 Conclusion and Future Work

Building information retrieval systems that efficiently and accurately respond to user queries is crucial. This paper introduces an additional intrinsic component—perspective—to the information retrieval process, ensuring that documents are not only relevant but also accurate. We present a new benchmark, PIR, which consists of six realistic scenarios designed to evaluate retrievers' sensitivity to different perspectives. Our experiments reveal biases in current retrieval technologies. To address this, we introduce a novel, zero-shot, projection-based method called PAP, designed to enhance perspective awareness without the need for further fine-tuning. Our analysis highlights the critical role of perspective awareness, demonstrating its impact on various applications. We hope the proposed PIR and PAP can benefit the community by developing retrievers with sensitive perspective awareness and hence ensure consistent performance in various downstream tasks.

We also observe potential future work to extend the scope of this paper: (1) Alignment between the retrieval and downstream task performance: as discussed in Section 3.3; (2) Perspective-aware re-ranking: in our paper, we only adapt the retrievers. In Section 3.4, we show that LLMs can identify perspectives in queries. However, using LLMs to identify the query-document relations will significantly increase the cost. Further methods should be designed to improve the efficiency of perspective-aware re-ranking, e.g., by identifying the necessary cases; (3) Perspective-aware instruction finetuning: TART achieves improved performance compared to the backbone Contriever. Our data creation pipeline and use of generative models can potentially help extract a natural collection of contrastive perspective-aware instructions in TART style. Further instruction-finetuning with such a collection is expected to improve retriever perspective awareness.

## Acknowledgments

The work was supported by the ONR Award N000142312840. The authors thank Xuanyu Zhou, Ruixin Hong, Vijay Viswanathan, Chenyang Yang, and Christina Ma for their valuable feedback, and anonymous reviewers for helpful discussions and comments.

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

# A   Appendix

## A.1   Dataset Examples

We present detailed examples of perspectives, root queries, and gold corpus candidates in Table 7. We can observe that different perspectives from the same task are commonly triggered by few-word differences, compared to the root queries. We can also observe the naturalness of root queries inherited from the original tasks and the diversity of involved components in style and semantics. Some details of the gold entries are omitted for better presentation.

| Dataset | Perspectives | Root Queries | Gold Corpus Candidates |
|---|---|---|---|
| AGNews | Find a news article related to Humanitarian crises in the same location of: 

 Find a news article that happened in Canada and has the same topic as: | An investigative analysis has found that India's lack of investment in public health infrastructure... | Investigative journalists have uncovered shocking evidence of extreme humanitarian crises in parts of India ... 

 In the wake of the ongoing COVID-19 pandemic, researchers in Canada have raised concerns about... |
| StoryAnalogy | Find a sentence that is an analogy to: 

 Find a story that is with similar entities with: | Fertilize the soil. Mix seeds into the fertilized soil. | Apply lotion to the skin. Massage the lotion into the skin. 
 Sprinkle hope into the nurtured soil, allowing dreams to take root and flourish. |
| Perspectrum | Find a claim that supports the argument: 

 Find a claim that opposes the argument: | It should be allowed to have military recruitment in schools | Young people should hear of the opportunities... 
 School children are too young to target for military service... |
| AmbigQA | in Nebraska 

 in Mississippi 

 in all but two states in the USA | What is the legal age of marriage, without parental consent or other authorization | In Nebraska, the legal age of marriage ... is 19 
 In Mississippi, the legal age of marriage ... 21 
 Most states share ... the legal age of marriage ... is 18. |
| AllSides | From Left-wing media, 

 From Right-wing media, | find a news article on the topic: terrorism | Story highlights Tsarnaev has said his brother drove the attack... 
 ...The twin bombings killed three people and wounded at least 176. Patrick told Fox... |
| Ex-FEVER | Find a claim that this sentence refutes: 

 Find a claim that this sentence supports: | Mother Teresa was born in Macedonia, a country in Europe where the residents are called Macedonians. Macedonian is a Slavic language. | Mother Teresa was born in a country in Europe where the residents called in a Slavic language. 
 Mother Teresa is a nun and missionary who was born in a country in Russia where the residents called in an Eastern South Slavic language. |

Table 7: Examples of the selected reformatted datasets. Root queries are queries without perspectives. Real queries in the datasets are not always the simple concatenation of perspectives and root queries. The root queries and gold corpus candidates demonstrated can be part of the actual ones in our benchmark.

## A.2   Details of Dataset Creation

The details of the involved dataset are as follows:

1. AGNews (Zhang et al., 2015; Yu et al., 2023b): we create a News Retrieval task from the synthesized AGNews generated by (Yu et al., 2023b). The queries for this task are finding a piece of similar news to the given news from a particular aspect: e.g., Find an article that happened in France and is similar to *News X*. (Yu et al., 2023b) leveraged a systematic approach to generating each piece of news with multiple control factors: length, location, subtopics, and styles. With this feature, we can reformat the dataset to a similar news retrieval task where each piece of query news only matches one target. We

choose topics (same style, same location, different topics) and locations (same style, same topic, different locations) as our perspectives.

2. StoryAnalogy (Jiayang et al., 2023): (Jiayang et al., 2023) create a large-scale dataset with human-annotated analogically similar story pairs, e.g., the *virus invades cells is analogically similar to burglars break in the house*. We sample the top 1,000 story pairs annotated with the highest analogical similarity and lowest semantic similarity in the original dataset. We create a story retrieval task with these pairs for the analogy perspective. To create similar stories from the semantic perspective, we leverage GPT-4 (OpenAI, 2023) to create similar sentences with respect to the query topic and entities.

3. Perspectrum (Chen et al., 2019): the authors of the original paper studied how evidence paragraphs from different sub-claims lead to different stances towards the original claims. We use the original claims as the queries of our retrieval task and leverage the human-annotated stances to form gold candidate evidence paragraphs from either *supporting, opposing* perspectives.

4. AmbigQA (Min et al., 2020): the authors of the original paper studied how the ambiguity of the questions leads to multiple plausible answers. Correspondingly, for each of the plausible answers, specific supporting evidence is attached. For example, for the question: *Who sings the song what a beautiful name it is?*, two perspectives are: *which group* and *who is the lead singer*. Answering these unambiguous perspective questions requires different supporting documents. We create our QA retrieval task in line with the general open-domain QA setting (Kwiatkowski et al., 2019). The original questions are the queries and corresponding supporting paragraphs from Wikipedia are the retrieval candidates.

5. AllSides (Baly et al., 2020): the original authors study how bias and ideology shape argument writing with news articles from different media and ideology annotation from a popular online forum[5]. For each topic, e.g., *is a 32-Hour Workweek a Good Idea?*, there are reports from multiple sources (e.g., *Fortune, Mew York Post*), with annotations from humans on whether the arguments are likely from the Left, Center, or Right-Wing ideologies. We choose the original topics as the questions and these ideologies as the perspectives. For AGNews, we create a similar news retrieval task. For AllSides, we create a topic-news retrieval task. We anticipate both scenarios to appear in the real world when users seek news articles.

6. Ex-FEVER (Ma et al., 2023): the original authors build a large-scale fact-checking dataset with multi-hop explanations attached to each of the claims. We treat the explanations as the queries for our Fact Retrieval task to study the relations among claims. Claims that explanations support, refute, or have no enough information as the retrieval targets. These retrieval targets are typically on the same topic and with one or two-word differences that make them either a piece of fact or not.

### A.3 Details of Baseline Retrievers

We present PIR as an evaluation benchmark for zero-shot information retrieval. We test the following zero-shot baselines to reveal the perspective awareness of different kinds of retrieval systems.

1. **BM-25**: We first compare with the traditional BM-25 retrieval method (Robertson & Zaragoza, 2009; Trotman et al., 2014), which is a popular key-word matching algorithm with TF-IDF token weights.

2. **Dense Passage Retrieval (DPR)**: DPR model (Karpukhin et al., 2020) was trained to retrieve the most relevant passage given a query with. In the experiment, we select a model trained with the natural question dataset as the representative. We treat all candidates (i.e., words/sentences/passages) as the passages and encode them with the pre-trained encoder.

3. **SimCSE**: To better measure the sentence similarities, developed over sentence encoder (Reimers & Gurevych, 2019), SimCSE (Gao et al., 2021) proposed to fine-tune

---

[5]AllSides: https://www.allsides.com/unbiased-balanced-news

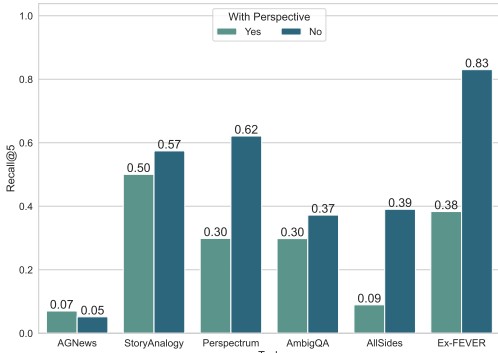 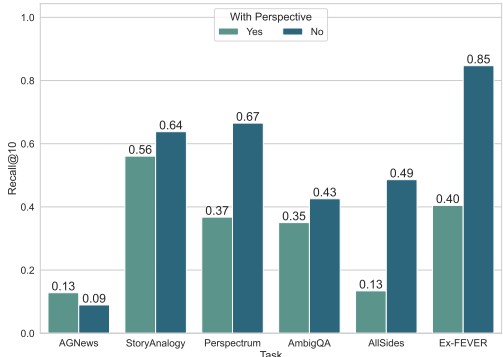

Figure 4: Retrieval performance (Recall@k, k=5,10) between queries with or without perspectives attached. Per-task performance is the macro-average of all the retrievers.

pretrained language models, e.g., BERT (Devlin et al., 2019), to capture the sentence-level semantics better. In the experiment, we use SimCSE as the representative sentence encoder, which shows good performance on retrieval tasks (Chen et al., 2023a). Specifically, we consider both the unsupervised (-unsup) and supervised (-sup) versions of SimCSE, where the former leverage dropout as signals and the latter fine-tunes the former with natural language inference data (Bowman et al., 2015).

4. **Contriever** (Izacard et al., 2022): Contriever is an unsupervised retriever, instantiated with a BERT-base encoder. (Izacard et al., 2022) leverage contrastive learning to fine-tune the encoder by segment pairs constructed from unlabeled documents from Wikipedia and web crawl data. Specifically, we use the Contriever model that is further trained on (MS-MARCO, Nguyen et al., 2016) for improved performance.

5. **TART** (Asai et al., 2022): TART is an instruction-tuned retriever that is fine-tuned on BERRI, an instruction-aware information retrieval dataset on approximately 40 tasks. The instructions are typically meta-data and descriptions about the corpus, e.g., retrieving a passage from Wikipedia. (Asai et al., 2022) show that involving such meta-data helps improve the retrieval performance, yet TART is not sensitive to wrong instructions provided. Similar to (Oh et al., 2024), we use TART-dual that is based on Contriever.

### A.4 Experimental Details

We collect datasets from the official GitHub repositories of the referred work. We conduct our experiments of retrieval on a machine with 8 Nvidia A6000 (40G) GPUs with CUDA 12 installed. For experiments on GPT-3.5-Turbo, we use the official API from OpenAI with a temperature equal to 1.0 and a max length equal to 80.

### A.5 Retrievers struggle to distinguish queries with perspectives

We explore how the retrievers perform with or without perspectives contained in each query for the PIR task. Ideally, the performance with perspectives shall be equal to or higher than the queries without perspectives (i.e., root queries) since additional information is provided.

We report the performance of queries with or without perspectives in Figure 4. The presented Recall@5 and Recall@10 are the averaged performance from all baseline retrievers. We can observe that, across different thresholds of recall, retrievers generally perform worse when the queries are with perspectives, which demonstrates that current retrievers cannot identify and follow perspectives to extract the correct corpus entries. This finding validates the significance of studying the problem of PIR.

One exception is the Recall@10 score from AGNews. One reason behind this can be that retrievers are biased towards News Articles from certain countries (e.g., Brazil) even when the overall similarity is low. In this case, providing perspective such as *find a news article from Egypt* can help.

| Retrievers | AGNews | StoryAnaloy | Perspectrum | AmbigQA | AllSides | Ex-FEVER | Avg. |
|---|---|---|---|---|---|---|---|
| no-intervention | 11.0 | 86.1 | 52.1 | 53.3 | 10.8 | 52.4 | 44.3 |
| add | 19.3 | 84.0 | 50.6 | 53.4 | 15.2 | 52.0 | 45.8 |
| add+ | **21.1** | 68.8 | 32.7 | 53.2 | 2.3 | 46.7 | 37.5 |
| concat. | 6.5 | 84.2 | 53.6 | 45.9 | **29.6** | **52.7** | 45.4 |
| concat.+ | 10.8 | 85.6 | 52.0 | 52.1 | 10.0 | 52.2 | 43.8 |
| cast | 4.6 | 83.3 | 51.7 | 43.5 | 26.4 | 52.6 | 43.7 |
| cast+ | 6.1 | 86.6 | 55.3 | 52.9 | 22.8 | 52.5 | 46.0 |
| dual-sum | 20.1 | 81.1 | 48.5 | 53.1 | 15.8 | 51.5 | 45.0 |
| trim-sum | 17.4 | 85.0 | 50.5 | 53.6 | 14.0 | 51.9 | 45.4 |
| PAP | 5.3 | 86.2 | **55.4** | 52.9 | 23.4 | 52.6 | 46.0 |
| PAP+ | 7.3 | **86.7** | 55.2 | **53.8** | 23.1 | 52.6 | **46.4** |

Table 8: Performance of different methods to improve the retriever perspective awareness on PIR, with $p$-Recall@5. as the metric. + denotes that the adaptation is also done on corpus entry embeddings.

| Method | SimCSE-base-sup | SimCSE-base-unsup | SimCSE-large-sup | SimCSE-large-unsup |
|---|---|---|---|---|
| PAP | 46.0 (+1.7) | 42.0 (+0.2) | 46.9 (+1.7) | 44.8 (+0.6) |
| PAP+ | 46.4 (+2.1) | 42.5 (+0.7) | 47.6 (+2.4) | 45.1 (+0.9) |

Table 9: Performance of PAP and PAP+ with different backbone models and scales. The improvement compared to the retriever without adaptation is in bracket.

### A.6 Full Results of Methods in Improving Perspective Awareness

In Table 4, we present the averaged $p$-Recall@5 for different zero-shot adaptation methods we perform. In Table 8, we present the task-specific performance. We can observe extra details from the results: PAP and PAP+ achieve consistent performance across tasks. However, they degrade the performance of AGNews. Yet *add*/*sum*-based methods achieve good improvement on AGNews. These findings suggest that building further taxonomy on the perspectives may help develop task-specific zero-shot adaptation. This can be an interesting future work beyond the scope of this work.

### A.7 Backbone and Scale Ablation

In this section, we test if PAP is robust with various backbones (supervised or unsupervised SimCSE) and backbone scales (SimCSE fine-tuned from BERT-base/-large). From Table 9, we can observe that we get consistent performance improvement using PAP and PAP+ to adapt the query and corpus embeddings. The gain is also higher with larger-scaled backbone SimCSE models.

### A.8 Details about Analytical Experiments

In this section, we describe the experimental details for downstream task performance (Section 3.3) and automatic perspective extraction (Section 3.4).

**Downstream Task Performance:** For AmbigQA, we use the prompt: ("Knowledge:{k}; Question:{q}; Answer:"). For essay writing with Perspectrum, we use the prompt ("You are writing a short argumentative essay (100 words) on a particular topic with a stance. We provide some background knowledge for your reference. Be concise and fluent; Background:{k}; Topic:{q}; Stance:{s}; Short Essay:"). For AmbigQA, we replace {k} with the top-1 retrieved entry. For Perspectrum, we replace {k} with the top-5 retrieved entries.

| Retriever | Perspectrum | | AllSides | | |
|---|---|---|---|---|---|
| | Support | Undermine | Left | Right | Center |
| INSTRUCTOR-base | 57.0 | 43.0 | 33.7 | 38.5 | 27.8 |
| INSTRUCTOR-large | 56.1 | 43.9 | 33.0 | 39.7 | 27.3 |

Table 10: Portion of different perspectives when the retrievers successfully retrieve relevant documents with root queries. We can observe that retrievers are biased towards supporting documents (for Perspectrum and Ex-FEVER) or news articles from the right-wing media (for AllSides). In the corpus, the number of entries related to each perspective is designed to be equal.

**Generative Perspectives from Queries:** Metric: Denoting the gold answer set as $\mathcal{G}$ and uni-gram tokens of each query and each corresponding gold answer as $\mathcal{T}_q$ and $\mathcal{T}_g$, the QA-F1 score can then be written as QA-F1 $= \max_{g \in \mathcal{G}}(\mathcal{T}_q \cap \mathcal{T}_g)/(\mathcal{T}_q \cup \mathcal{T}_g)$.

Prompt Template: the prompt template we use is: "You are doing a task to reformat queries into input for a retriever. here are two components in the query: a short phrase describing the user intent perspective and the rest describing the general information; Can you decompose the query and find the intent perspective for me? Answer with part of the query only, no other information is needed; Query: {q}; Intent:"

### A.9 Other Instruction-tuned Sentence Embeddings

(Su et al., 2022) introduce a method to involve instruction in text embeddings with fine-tuning on a mixture of 330 multi-purpose tasks. The instructions are general descriptions of what the task is about, e.g., Represent the caption for duplicate captions.

The extraction of INSTRUCTOR-style embeddings is different from the general text embedding methods we introduced in Section 3.1, which makes them not fairly comparable. The model $\phi_{\text{instructor}}$ requires both an instruction and a piece of text as the input. We treat perspectives as instructions here and root queries as the text input. The model design involves the decomposition of perspectives and root queries by default. We test both INSTRUCTOR-base and INSTRUCTOR-large.

We follow our setting in Section 2.5 to test if the INSTRUCTOR is also biased towards certain perspectives. From Table 10, we can observe that the gap still remains and instruction fine-tuning on general tasks does not solve the problem of perspective awareness.

### A.10 Details of retrievers can be socially biased towards certain perspectives

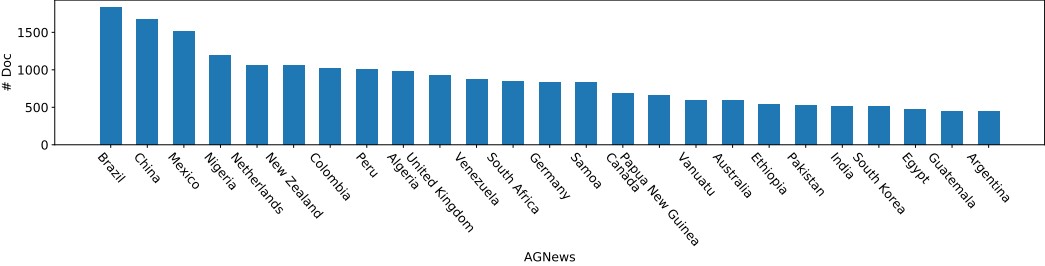

Figure 5: Accumulated numbers of news articles the top 5 retrieval results with SimCSE-sup on AGNews root queries. An example of the root query is: Find a news article that is similar to: Article X. We can observe that retrievers prefer news articles from certain countries, e.g., Brazil. In the corpus and root queries, the numbers of articles per country are designed to be equal.

In Section 2.5, we show how retrievers are biased from certain perspectives, which can degrade the downstream task performance. On the other hand, users seeking news articles from different countries may expect different rates of success.

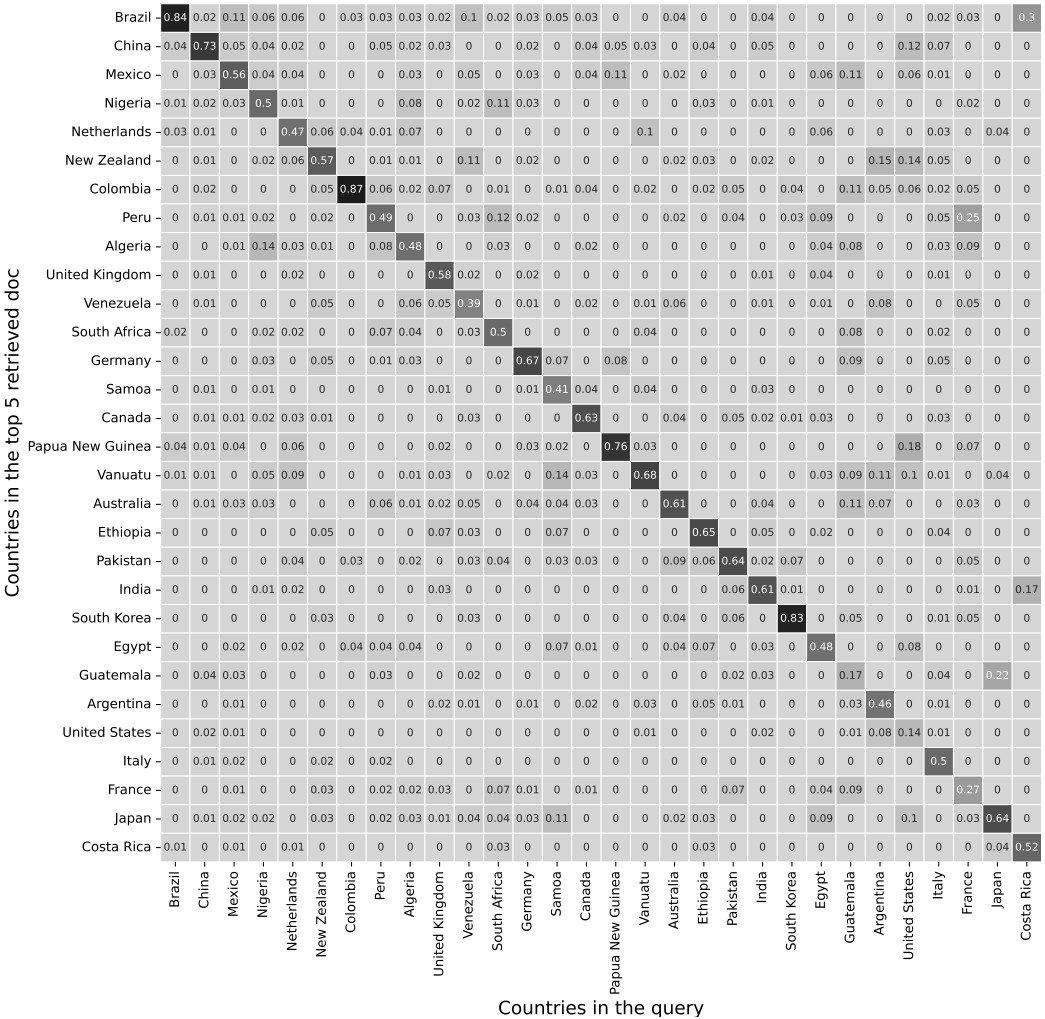

Figure 6: Expected portion of news articles from the desired or other countries in the top 5 retrieval results with SimCSE-sup on AGNews. Queries are from the location perspective, e.g., Find a news article on X topic and happen in Y, where Y is the desired country.

In this section, we provide more details of the analysis of social bias by answering two questions:

1. if the queries do not contain a desired country, will the bias still exist? Figure 5 shows that the bias still exists, news articles from countries such as Brazil are over-represented.

2. What are the actual countries in *News article from other countries* in Figure 3? We unfold Figure 3 and create a heatmap in Figure 6. We can observe that, similar to Figure 5, news articles are over-represented even when the desired origin countries of the articles present in the queries.

