# OpenReview forum: "Beyond Relevance: Evaluate and Improve Retrievers on Perspective Awareness"
_colmweb.org/COLM/2024/Conference — COLM_

### Official Review · Reviewer_CcDN · 2024-05-08

**Rating:** 7
**Confidence:** 4
**Ethics Flag:** 1

**Summary:**

This work introduces an additional intrinsic component—perspective—to the information retrieval process and presents a new benchmark, PIR. It also introduces a zero-shot, projection-based method called PAP. Experimental results show that current retrievers have limited awareness of subtly different perspectives in queries and can also be biased toward certain perspectives. Meanwhile, the method PAP outperforms other baseline methods in multiple settings and can enhance the perspective sensitivity of the retrieval process in a zero-shot manner without additional fine-tuning. Furthermore, perspective-aware retrieval can significantly improve the performance of downstream tasks.

**Questions To Authors:**

1. Regarding Weakness 1:
-It is unclear whether the terms 'root queries' and 'neutral queries' used in the abstract refer to the same concept. Could you please clarify?
-In the concluding sentence of Section 2.1, is the 'q' in "For each q" intended to represent the tuple (p, r)? Does this notation imply both an implicit 'p' within 'q' and an extracted 'p'?
-Are the formulations 'q = (p, r)' and 'q = p + r' mentioned in Section 2.1 and Equation 1 consistent with their respective descriptions in Table 4?
-What is denoted by 'S' in the final sentence of Section 2.3?
-Could you define what is meant by 'knowledge part' and 'gold knowledge' as referenced in the first sentence of the third paragraph of Section 3.3?
2. The study primarily investigates zero-shot information retrieval (IR). Could you speculate on the potential outcomes or adjustments necessary for few-shot IR scenarios?

**Reasons To Accept:**

1.The underlying motivation for this research is compelling and well-founded.
2.The methodology introduced is both straightforward and efficacious.
3.This study establishes a new benchmark that is thoroughly examined and analyzed in detail.

**Reasons To Reject:**

1.Certain sections of the text lack clarity, and the logical progression appears flawed. We will address specific questions regarding these issues in the Questions section.
2.While Section 3.4 employs GPT-3.5 to generate perspectives for QA downstream tasks, experiments do not extend to retrieval tasks. It would be advantageous to include retrieval experiments using perspectives generated by GPT-3.5 in Table 4 as well.
3.The manuscript lacks a comparative analysis with existing research on cosine similarity. It would be beneficial for the authors to conduct comparative experiments with the methodologies outlined in https://arxiv.org/abs/2403.05440.

---

> ### Author Rebuttal · Authors · 2024-05-30
>
> Thank you for your valuable and detailed feedback. We would like to address your questions as follows.
>
> **Q1: Notations**
>
> Thank you for the detailed suggestions on the notations. We will update the final version accordingly.
>
> 1. Neutral describes the root queries in the PIR task, which is contextually used to describe the non-perspective part of the user query.
>
> 2. q = p+r  is q = (q,r). We will unify the notation and add an explanation.
>
> 3. S is a typo of the corpus C.
>
> 4. The knowledge part refers to the prompt in Appendix Section A.8 used for the experiments in Section 3.3, as mentioned in the last sentence of the second paragraph in Section 3.3.
>
> **Q2: Generative Perspectives for Retrieval**
>
> Thanks for the suggestions. We extend the experiment with the generative perspectives (50 queries per task, full corpus) on retrieval tasks to test the performance consistency, with Recall@5 as the metric. For each retriever (e.g., BM25), A - B denotes the retrieval performance with annotated perspectives (A) and generative perspectives (B)
>
> | Dataset | QA-F1| BM25 | DPR | SimCSE-sup |
> | :---: | :---: | :---:| :---:| :---:|
> AGNews | 78 |12 - 12 | 4 - 4 | 8 - 8 |
> StoryAnalogy | 66 | 56 - 60 | 26 - 32 | 36 - 38 |
> Perspectrum |	69 | 26 - 28 | 38 - 38 | 50 - 50 |
> AmbigQA | 70 | 21 - 18 | 46 - 44 |44 - 44 |
> Allsides | 66 | 6 - 8 | 8 - 10 | 8 - 8 |
> Ex-FEVER| 69 | 80 - 82 | 60 - 60 | 74 - 72 |
>
> From the table above we can observe that the retrieval performances from annotated and generative perspective extraction are similar across tasks and retrievers, which validates our claims in Section 3.4.
>
> **Q3: Existing research on cosine similarity**
>
> The authors indeed read the referred paper and found it interesting. It was put on ArXiv on March 8 and would be considered concurrent work under COLM policy, but we still intend to add a discussion in the final paper: We use cosine similarity for fair comparison since current retrievers such as DPR and Contriever are tuned on similarity from the cosine similarity function.
>
> **Q4: Few-shot IR**
>
> We speculate that the proposed PAP method can benefit from the few-shot IR setting. As in our discussion with Reviewer 8C7L, we can design a tunable parameter to decide how much perspective awareness to be involved per query, which can be tuned with the shots given to perform task-specific PAP.
>
> Another potential benefit is to find the task-specific perspectives to improve computational efficiency, as described in Section 3.1.

---

> > ### Author Response · Authors · 2024-06-04
> > **Looking Forward to Further Discussion**
> >
> > Dear reviewer CcDN,
> >
> > We would like to thank you again for the feedback and suggestions. We hope our clarification and results on the setting you mentioned addressed your concerns. We are committed to improving our manuscript and would appreciate it if you find our response adequate and consider raising the score.

---

> > ### Comment · Reviewer_CcDN · 2024-06-06
> >
> > Thank you for replying to my questions.
> >
> > - After considering the author's rebuttal, I have decided to revise the score and increase it to 7 points.
> > - I believe the author's experiment answered my question well and is convincing.

---

> > > ### Author Response · Authors · 2024-06-06
> > > **Thank you for your review**
> > >
> > > Dear reviewer CcDN,
> > >
> > > Thank you so much for your review and for liking our paper. We will update the paper with the points we discuss and put continuous effort into improving the perspective awareness of retrievers.

---

### Official Review · Reviewer_K88D · 2024-05-11

**Rating:** 6
**Confidence:** 3
**Ethics Flag:** 1

**Summary:**

The three main contributions of this paper are (1) the introduction of a perspective-aware information retrieval (PIR) benchmark by repurposing existing retrieval benchmarks, (2) a method for improving perspective-aware retrieval by projecting queries (and potential the datastore) into perspective space (PAP), and (3) an empirical analysis of off-the-shelf retrievers on PIR and how to build in PAP.

**Questions To Authors:**

1. Bottom of page 6: "PAP variances" => variants
2. The intent perspective extraction prompt in appendix A.8 is simple, and I would expect "intent perspective" to be a rare bigram. I am therefore surprised the performance is high for generating perspectives from queries. Do you have any intuition for why this works?
3. The improvement over the no-PAP retrieval baseline in Perspectrum is large, but the improvement over the no retrieval baseline is small. Can you explain this?

**Reasons To Accept:**

* The paper is well-written and the experiments are thorough.
* The proposed problem and benchmark are timely, as many works are concurrently identifying shortcomings in retrieval methods.
* The perspective projection method is also simple and intuitive.

**Reasons To Reject:**

The empirical improvement from incorporating the PAP project method is modest, both for recall and downstream tasks. My stance is that the authors have not found the best downstream task to really highlight their method and that this paper is one or two experiments away from being a strong paper. Perspectrum seems to be on the right track -- I suspect the downstream tasks that will benefit the most from perspective-aware retrieval will be reasoning-intensive (rather than knowledge-intensive) and require comparing multiple perspectives. One could imagine trying to prove a theorem by retrieving lemmas with different perspectives, e.g. geometric or algebraic.

I am open to being convinced that the improvement in downstream tasks is significant.

---

> ### Author Rebuttal · Authors · 2024-05-30
>
> Thank you for your valuable and detailed feedback and for liking our paper. We would like to address your questions as follows.
>
> **Q1:  Downstream Tasks**
>
> We sincerely thank you for the insights into the choice of downstream tasks. In this paper, we focus on introducing the problem and designing the benchmark. We start with zero-shot PAP to improve current retrievers on the retrieval and downstream tasks that are currently within our included datasets (i.e., AmbigQA and Perspectrum).
>
> We believe continuous effort shall be made to improve retrievers’ perspective awareness (e.g., add tunable parameters and mine automatic signals). We will apply the retrievers with high perspective awareness (which PIR can reveal) to other tasks. We will add a section describing the potential application to other tasks as a roadmap for the authors and other researchers, for example, TheoremQA [1]. We will also work on including these tasks in the PIR benchmark.
>
> **Q2:  Writing details**
>
> Thanks for pointing out the typos and writing comments. We will update them in the final paper. For the intuition on other questions:
> One potential reason is that the LLMs can understand “two components” (where one is the “intent”) in the prompt in Appendix A.8. As shown in examples in Section A.1, the user meta-description (e.g., “Find a news article”) and general information (e.g., the source news content) can be easy to distinguish. We are experimenting with a zero-shot setting. We believe demonstrations can further stabilize the performance. We are also open to all prompting suggestions.
>
> The large improvement over the no-PAP retrieval baseline is also interesting to the authors. We believe this is rooted in the decoupled relation between the retrieval and downstream task performance. The retrievers may find relevant but incorrect documents, which may hurt the instruction-following LLM performance compared to no-retrieval. We introduce some related readings in the last paragraph of Section 3.3. Together with these related papers, this improvement mismatch suggests our takeaway: beyond relevance that makes a retriever successful on retrieval tasks, we should be careful of the correctness of responding to the perspectives of user queries to make the whole RAG pipeline successful.
>
> [1] Wenhu Chen, Ming Yin, Max Ku, Elaine Wan, Xueguang Ma, Jianyu Xu, Tony Xia, Xinyi
> Wang, and Pan Lu. 2023. Theoremqa: A theorem-driven question answering dataset. In EMNLP 2023.

---

### Official Review · Reviewer_HLF6 · 2024-05-12

**Rating:** 6
**Confidence:** 3
**Ethics Flag:** 2

**Summary:**

This paper explores a new retrieval setting in which different perspectives of the queries are to be recognized and attended to by the retrieval system. The key requirement is that the retrievers should attempt to address the subtle different perspectives expressed in the queries. A new benchmark for the task (Sec 2) is created, including a new variant of the recall metric (Eq 1) that takes perspectives into account. A new approach is proposed to improve the perspective awareness of a SimCSE baseline, called perspective-aware projection (Sec 3), by performing a perspective/corpus projection over the input embeddings. The main results in Table 5 show that the proposed methods can improve the accuracy and correlation of SimCSE on AmbigQA and Perspectrum datasets respectively by a wide margin.

This paper is written up carefully, has a clear storyline regarding a key limitation in mainstream retrieval paradigm about perspective fairness. It shows that while SimCSE as a retriever doesn't always do a good job in a perspective-awareness scenraio (Table 5), this side of effectiveness can be further enhanced by employing a geometric projection approach PAP and PAP+. The overall narratives and the experimental effort are interesting and insightful, the methodology is simple and easy to reproduce.

I'm not generally across the fairness line of work for retrieval, but the main point raised in the paper ("improving perspective awareness") seems crucial and timely. Technical novelty aside, all three main claims are reasonably demonstrated and fulfilled in separate sections. I think the paper can meet the bar for acceptance and be included in the proceedings.

**Questions To Authors:**

- In Sec 2.2, you write "we use heuristics to ensure that all the queries..." Can you make unpack this mention of heuristics and call out explicitly what you do in the main text?
- Eq 1 seems to use a cyclic dependency: $q = p + r$ but then $p$ is to be sampled from a set conditioned on $q$.

**Reasons To Accept:**

- This paper presents a new formulation about perspective awareness and some practical dataset and methodology for evaluation. Given that LLMs are now used widely in all sorts of applications, this type of evaluation effort that strengthens model safety and reliability is particularly timely.
- The paper shows that this new perspective-awareness evaluation criterion is not unreasonably difficult. A geometric projection method can be applied to SimCSE and improve the effectiveness.

**Reasons To Reject:**

- On the methodology side, Sec 3 only reviews a simplistic geometric approach. I do not believe technical novelty would be a limiting factor for this type of work, but given that ranking/prediction fairness has been studied for many years in NLP & IR there should be some relevant prior art that we can refer to and compare the proposed method with.

---

> ### Author Rebuttal · Authors · 2024-05-30
>
> Thank you for your valuable and detailed feedback and for liking our paper. We would like to address your questions as follows.
>
> **Q1: Beyond geometric approach**
>
> Thank you for the valuable insights on the potential methods. In this paper, we focus on introducing the problem and designing the benchmark. We believe continuous effort shall be made to improve retrievers’ perspective awareness. As also mentioned in our response to Reviewer 8C7L, one direction is introducing a tunable parameter to decide how much perspective awareness should be involved per query.
>
> With the motivation from retrieval diversity work [1][2], we also work on how to automatically mine and generate diverse and contrastive retrieval signals to help train retrievers with high perspective awareness. We will extend the future work discussion in current Section 5 in the final paper.
>
> **Q2: Writing details**
>
> For the heuristics in creating the datasets, we introduce how we leverage and extend the design of source datasets to create the query-document pairs in Section A.2 in the appendix, with examples in Table 7. We will include a detailed introduction of the key reasons and unpack the mention in the final version.
>
> The dependency issue is from the difference in the cases in the wild (general case) and in our dataset design (PIR). In the general case, retrievers only observe query (q), external tools can be applied to sample perspectives (p) and root queries (r) from q, such as the generative perspectives we describe in Sec 3.4
>
> In PIR,  to test the model perspective awareness, we design the dataset to have multiple p attached to each r. so that in the metric related to PIR (i.e., in Section 2.3), we write q = p+r, specifically for the PIR task. We will add an explanation of the setting differences in the final paper.
>
> 𝑞=𝑝+𝑟 describes the construction of our PIR dataset: To ensure that the perspectives are explicit and have contrastive pairs, we first sample perspectives (p) from existing documents (such that we know the gold document to retrieve for each p), then add them to a root query (r), such that r+p forms the final query (q) submitted to the retriever. In real-world scenarios, p and r tend to be implicit and can only be decomposed from q (the only observable data point) – a case we discussed such decomposition in Sec 3.4. We will explain the equation in more detail.
>
> [1] https://plg.uwaterloo.ca/~gvcormac/novelty.pdf
>
> [2] https://ieeexplore.ieee.org/document/7987038

---

> > ### Comment · Reviewer_HLF6 · 2024-06-03
> >
> > Thank you for replying to my questions.
> >
> > - Happy for you to take the initiative to address these limitations outright or outline an agenda for them to be looked at in future. I think as a scientific exercise it's interesting to include some relevant comparisons even if they're outside the scope initially.
> > - I'm not entirely sure about this loopy dependency but if fixing the notation (and descriptions as well) could improve clarity please by all means do so.

---

> > > ### Author Response · Authors · 2024-06-03
> > > **Thank you for the reply**
> > >
> > > Thank you for the reply. Along with our ongoing work on improving the retriever perspective awareness, we will also write a clear agenda and provide open-source code support in the final paper. We will fix the notation issue and extend the task formulation discussion in Section 2.1. Thanks again for liking our paper.

---

### Official Review · Reviewer_8C7L · 2024-05-14

**Rating:** 6
**Confidence:** 4
**Ethics Flag:** 1

**Summary:**

This paper formally introduces the task of perspective-aware retrieval, that conditions retrieval upon a specific perspective (e.g. opposing or supporting a claim). This has important influence over downstream tasks, such as RAG. The paper introduced a benchmark for this new task, and designed a simple method that outperforms various baselines.

**Questions To Authors:**

- Section 3.1 the PAP method: the equation provided moves the query vector away from the perspective vector by some amount, it does not project it onto the p space. Additionally, in the term prefixed with the minus sign, there could be an coefficient parameter to tune. Please explain.
 - Table 4: $\cos(q, c)$ is not a correct notation since $\cos$ takes 1 parameter. Please write $\cos\angle(q, c)$. The `dual-sum` and `tri-sum` lacks intuition to me.
 - Section 2.5: you argue that SimCSE, not specifically trained with any specific task outperforms. However, you stated that an NLI-fine-tuned SimCSE performs better. Please clarify.

**Reasons To Accept:**

The paper is very clearly written and easy to understand, and has important impact over downstream tasks such as RAG.

**Reasons To Reject:**

The mathematics of the projection method seems dubious, see questions below.

---

> ### Author Rebuttal · Authors · 2024-05-30
>
> Thank you for your valuable and detailed feedback and for liking our paper. We would like to address your questions as follows.
>
> **Q1: Details about the PAP method**
>
> Thank you for your valuable insights on the methodology side. By projection to the p space, we mean the hyperplane that is vertical to the perspective (p) vector. Reducing the query vector component on the perspective vector is equivalent to moving it toward the hyperplane (in the sense of cosine similarity). We will update the introduction of PAP in Section 3.1 with your suggestions.
>
> We agree that a tunable coefficient is interesting and can help reveal the essence of the problem (relations between perspectives and the rest of the query). In the scope of the current draft, we focus on introducing the task and building zero-shot methods. However, we are indeed exploring the coefficient tuning and will update the resources in our code base to be publicized upon acceptance.
>
>  **Q2: Notation and Baseline**
>
> Thanks for the suggestion on the notation. We will improve the final version as suggested.
> We design dual-sum and tri-sum for the comprehensiveness of possible operations on the similarity score level, while others, such as add and concatenation, are operations on the vector/embedding level.
>
> **Q3: Clarification on SimCSE**
>
> SimCSE is a sentence embedding model with two versions: the unsupervised one is fine-tuned with zero-shot contrastive signals from dropouts; the supervised one is further fine-tuned on NLI tasks. We mean that  SimCSE is not specifically fine-tuned on retrieval tasks (In contrast, the Contriever model used in our paper is further tuned on MS-MARCO, a retrieval task). We will add a detailed explanation in the final version.

---

### Decision · Program_Chairs · 2024-07-10

**Decision:**

Accept

**Comment:**

The paper defines a benchmark for perspective-aware retrieval by processing six existing IR datasets into a format of queries separated into a root and perspective portions, and corresponding relevant documents. The authors study the performance of 5 off-the-shelf retrievers and analyze accuracy for the generic and perspective-aware tasks, indicating interesting inaccuracies and biases of the models.

The authors also propose a method to derive query and document embeddings from off-the-shelf retrievers to make the representation more perspective-aware. Even though the transformations are not intuitive, they lead to improvements in average performance, as measured on retrieval metrics and end-tasks of retrieval-augmented generation for two tasks on a small number of examples. The improvements are larger in the end tasks although the sample size is small and the evaluation on one of the tasks is a bit unconvincing, as it relies on an automatic measure of sentiment.

Pros
 * the paper focuses on an important problem
 * the results based on the new datasets are interesting and provide insights into directions for future improvement of retrieval systems
 * the presentation is good, except for the motivation of the query/document projection method
 * the authors have promised to fix the notation in the mathematical development and added new useful results based on the reviewer comments

Cons
  * the corpora used for the retrievers are very small, with the largest one having 2900 documents
   * the representation projection method and its performance are not analyzed very well
   * manual evaluation would be more convincing for the perspective-aware Perspectrum RAG downstream task